# The Impact of Tumor Boards on Breast Cancer Care: Evidence from a Systematic Literature Review and Meta-Analysis

**DOI:** 10.3390/ijerph192214990

**Published:** 2022-11-14

**Authors:** Andrea Di Pilla, Maria Rosaria Cozzolino, Alice Mannocci, Elettra Carini, Federica Spina, Francesco Castrini, Albino Grieco, Rosaria Messina, Gianfranco Damiani, Maria Lucia Specchia

**Affiliations:** 1Clinical Governance Department, Fondazione Policlinico Universitario A. Gemelli IRCCS, Largo Agostino Gemelli 8, 00168 Rome, Italy; 2Department of Life Sciences and Public Health, Università Cattolica del Sacro Cuore, Largo Francesco Vito 1, 00168 Rome, Italy; 3Azienda Regionale Emergenza Sanitaria 118, Via Portuense 240, 00149 Rome, Italy; 4Faculty of Economics, Università “Universitas Mercatorum”, Piazza Mattei 10, 00186 Rome, Italy; 5ASL Roma 1, 00193 Rome, Italy; 6Department of Maternal, Children and Adult Medical and Surgical Sciences, Università degli Studi di Modena e Reggio Emilia, 41121 Modena, Italy; 7Women, Children and Public Health Sciences Department, Fondazione Policlinico Universitario A. Gemelli IRCCS, Largo Agostino Gemelli 8, 00168 Rome, Italy

**Keywords:** tumor board, multidisciplinary team, breast cancer, healthcare, outcomes, diagnosis, treatment, survival

## Abstract

Breast cancer is the most common malignancy in women, with a complex clinical path that involves several professionals and that requires a multidisciplinary approach. However, the effectiveness of breast cancer multidisciplinary care and the processes that contribute to its effectiveness have not yet been firmly determined. This study aims to evaluate the impact of multidisciplinary tumor boards on breast cancer care outcomes. A systematic literature review was carried out through Scopus, Web of Science and Pubmed databases. The search was restricted to articles assessing the impact of MTB implementation on breast cancer care. Fourteen studies were included in the review. The most analyzed outcomes were diagnosis, therapy and survival. Four out of four studies showed that, with implementation of an MTB, there was a change in diagnosis, and all reported changes in the treatment plan after MTB implementation. A pooled analysis of three studies reporting results on the outcome “mortality” showed a statistically significant 14% reduction in mortality relative risk for patients enrolled versus not enrolled in an MTB. This study shows that MTB implementation is a valuable approach to deliver appropriate and effective care to patients affected by breast cancer and to improve their outcomes.

## 1. Introduction

Breast cancer is the most common malignancy in women [1]. In 2020, there were 2.3 million women diagnosed with breast cancer and 685,000 deaths globally. At the end of 2020, there were 7.8 million women alive who had been diagnosed with breast cancer in the past five years, making it the world’s most prevalent cancer [2]. The risk factors include age, reproductive factors, hormonal factors, dietary and metabolic factors, previous thoracic radiotherapy, previous dysplasia or breast cancer, family history and genetic predisposition [3]. Due to screening and greater awareness amongst women, most breast malignancies are diagnosed at an early stage when surgical treatment can more often be conservative and the therapy adopted more effective, with, in consequence, higher five-year survival rates [4].

As with other cancers, breast cancer treatment requires a highly complex approach, and quality care is dependent on coordinated multidisciplinary input [5,6]. 

The path of the patient diagnosed with neoplasia is complex and involves several professionals, from the surgeon to the psychotherapist. In this context, structured healthcare is necessary to improve outcomes and promote the coordinated management of the patient’s care. Therefore, to pursue quality objectives in healthcare, a multidisciplinary approach has become essential and represents the current trend in the United Kingdom, Europe, the United States, Asia and Australia [7]. Multidisciplinarity represents an organisational response to the need to make increasingly complex clinical decisions and, consequently, represents an essential aspect of oncological treatment. Moreover, many scientific studies reported in the literature describe multidisciplinary tumor boards (MTBs) as the therapeutic standard in oncology [8]. MTBs consist of multidisciplinary teams composed of different clinical specialists working together to make shared decisions regarding the clinical pathway of cancer patients. This approach ensures that all professionals involved in the oncological clinical pathway take part in the process of diagnosis, planning, assessment and treatment [9]. MTBs have also been reported to have an impact on cancer care, in terms of change to treatment strategies and processes, clinical results improvement, reduction in clinical outcomes variability, better use of economic resources and improvement in healthcare personnel training [10].

With respect to breast cancer, in many countries, multidisciplinary care has been formally established as an essential practice in its management and has been used as a benchmark for accreditation and funding [11]. Research into multidisciplinary breast cancer care varies by design, clinical context and study outcomes considered [12]. According to J. Shao et al., 2019, there is growing emphasis on the application of multidisciplinary approaches to breast cancer care. However, because of heterogeneous definitions and contexts, the effectiveness of multidisciplinary care and the processes that contribute to its effectiveness have not yet been determined with certainty [12].

Starting from an umbrella review published by Specchia ML et al. in 2020 [9], regarding the impact of MTBs on oncological patients’ care in general, the present systematic review aims to evaluate the impact of MTBs on breast cancer care outcomes. 

## 2. Materials and Methods

A systematic literature review was performed based on a structured search process using a specific search algorithm, an accurate study selection process, data extraction and quality assessment of the included studies. 

A systematic search of studies was carried out through Scopus, Web of Science and Pubmed databases using the following search string: (“breast neoplasms”[Mesh] OR “breast cancer*” OR “breast neoplasm*” OR “breast tumor*” OR “breast carcinoma”) AND (“patient care team”[Mesh] OR “patient care team*” OR “tumor board*” OR “multidisciplinary team*” OR “multidisciplinarity” OR “MDT” OR “multidisciplinary team meeting*” OR “multidisciplinary meeting*” OR “cancer conference*” OR “multidisciplinary care conference*” OR “MCC” OR “cancer meeting*” OR “healthcare team*” OR “interdisciplinary health team*” OR “clinical board*” OR “multidisciplinary treatment” OR “multidisciplinary management” OR “multidisciplinary care” OR “teamwork” OR “team work”) AND (“outcome*” OR “process” OR “assessment” OR “evaluation*” OR “efficacy” OR “efficiency” OR “effectiveness” OR “cost*” OR “caseload” OR “workload” OR “plan” OR “planning” OR “decision making” OR “impact” OR “personal*” OR “precision medicine”).

The review considered all studies published from 1995 to 28 June 2021. This year range was set since the multidisciplinary approach in cancer care was not regularly implemented in clinical practice until the late 1990’s, more specifically, from 1997, though it has been described in the literature from 1975 [9]. The retrieved studies were independently assessed by two researchers by first reading the title, and, if deemed suitable, then reading the abstract, before finally reading the full text based on the inclusion and exclusion criteria described below. Uncertainties arising regarding the inclusion of articles were overcome by discussion amongst the research members.

Inclusion and exclusion criteria were defined according to the PICO (Population, Intervention, Comparison, Outcome) model as follows: (P) The studies considered were focused on breast cancer patients; (I) The intervention was represented by multidisciplinary teams in a hospital setting. A multidisciplinary team was defined as a team composed of professionals from different clinical backgrounds who make decisions together to recommend the clinical pathway of an individual patient [9]; (C) The comparison was with lack of MTB implementation (patients’ clinical management without the intervention of multidisciplinary teams); (O) Both clinical outcomes (e.g., measurable change in symptoms, overall health, quality of life, or survival/mortality) and process-related outcomes (regarding the decisions made about the clinical journey in terms of diagnosis and therapy) were taken into account. The search was restricted to articles assessing the impact of MTB implementation, written in the English or Italian language, published between 1995 and 2021, and for which full text was available. Reviews, letters and commentaries were excluded because they do not report original data. 

Articles were selected according to the PRISMA statement criteria and the data were extracted independently by two researchers. For each study the researchers retrieved and summarized data related to the following: authors, year of publication and country of origin, study population, study design, method/intervention, main outcomes, main findings and quality assessment. 

To assess the quality of the included studies the Quality Assessment Tool for Observational Cohort and Cross-Sectional Studies and the Quality Assessment Tool for Before-After (Pre-Post) Studies With No Control Group [13] from the National Heart, Lung and Blood Institute Study Quality Assessment Tools were used. The quality assessment was carried out by two researchers independently with disagreements discussed until consensus was reached. The Quality Assessment Tool for Observational Cohort and Cross-Sectional Studies consists of fourteen questions with three possible answers for each item: ‘Yes’, ‘No’ or ‘Other’ (cannot determine, not applicable, not reported). Observational studies were labeled as of ‘Good’, ‘Fair’, or ‘Poor’ quality based on the cut-off points assigned: ‘Good’ quality (above 9 points), ‘Fair’ quality (between 9 and 5 points), ‘Poor’ quality (below 5 points). The Quality Assessment Tool for Before-After (Pre-Post) Studies With No Control Group consists of twelve items and the possible answers for each item are: ‘Yes’, ‘No’ or ‘Other’. With the same methodology, based on points assignment, before-after (pre-post) studies were labeled as of ‘Good’ quality (above 8 points), ‘Fair’ quality (between 8 and 4 points), or ‘Poor’ quality (below 4 points).

To combine non-homogeneous results, when appropriate (at least two studies reporting comparable outcomes), a meta-analysis was performed. Data regarding the detailed inclusion criteria, duration of follow-up and outcome (adjusted hazard risks (HRs and 95% confidence intervals (CIs)) for patients enrolled and not enrolled in MTBs) were extracted from each individual study. Study-specific estimates were combined using inverse variance-weighted averages of logarithmic HRs in both fixed and random-effects models (primary meta-analysis). Heterogeneity was assessed by Chi^2^ and I^2^ test. If no significant heterogeneity was observed (I^2^ ≤ 50%, *p* > 0.1), a fixed-effects model was adopted; otherwise, a random-effects model was applied [14]. Meta-analysis was performed using the Review Manager software (version 5.4.1).

## 3. Results

A total of 8998 records were retrieved from the search, of which 5303 were on Pubmed, 2014 on Scopus and 1691 on the Web of Science. After removal of duplicates, the final number of records retrieved was 6658. Selection by title and abstract reduced the total to 71 studies that were assessed for eligibility for reading the full text. Of these, 14 studies, satisfying the stated inclusion/exclusion criteria, were eligible for the study and were included in the review. A PRISMA flowchart provides details of the study selection process (Figure 1).

The studies included were published between 2006 and 2021 [5,8,15,16,17,18,19,20,21,22,23,24,25,26].

Four out of 14 studies (29%) were from the USA [5,15,22,24] and three (21%) from the UK [16,17,21]. The remaining seven studies (50%) were from seven different countries (one from Canada [18], one from Lebanon [19], one from Australia [20], one from China [8], one from Taiwan [23], one from Romania [25], and one from Mozambique [26]. 

The sample size was from 20 [25] to 61039 patients [5]; in 50% of the studies the sample size was between 149 and 657 [15,17,19,20,21,24,26]. 

Regarding the study design: six (43%) were historical cohort studies [5,8,20,23,24,26], five (36%) were before and after studies [15,18,19,21,22], one (7%) was a cohort study [25], one (7%) was a retrospective interventional cohort study [16] and one (7%) was an observational retrospective study [17]. 

Within the selected articles, the following outcomes were identified: diagnosis [15,18,19,22], therapy [5,15,18,19,20,21,25], survival [8,26], mortality [16,20,23], recurrence [23], time from visit to chemotherapy/surgery [24], prophylactic mastectomy [17], physical recovery [25], therapeutic compliance [25] and MTB cost-effectiveness [26].

The most analyzed outcomes were therapy and diagnosis, evaluated in seven (50%) [5,15,18,19,20,21,25] and four (29%) [15,18,19,22] studies, respectively.

Eight out of 14 studies (57%) analyzed more than one (from two to three) outcome [15,18,19,20,23,24,25,26] and, of these, three studies [15,18,19] analyzed both diagnosis and therapy outcomes.

In line with the search string, all the studies assessed the impact of MTBs on breast cancer care. The findings are detailed below.

### 3.1. Diagnosis

In terms of diagnosis, four out of four studies showed that, with implementation of the MTB, there was a change in this outcome [15,18,19,22]. In particular, three studies reported a change both in imaging and anatomo-pathologic interpretation [15,18,22].

### 3.2. Therapy

Changes in the treatment plan after MTB implementation were reported by seven out of seven studies [5,15,18,19,20,21,25]. With respect to the surgical treatment plan, as reported by Foster et al., 2016, patients followed by the MTB received a treatment recommendation different from that previously proposed by a single professional [18]. In one study [20], patients discussed at the MTB were found to undergo surgery significantly more than patients not discussed. A statistically significant increase in reconstruction was also highlighted [21].

### 3.3. Mortality

In two studies, the implementation of the MTB was associated with a reduced risk of mortality [16,23], while in one study [20], no statistically significant difference in mortality was found between patients discussed and patients not discussed.

A pooled analysis of three studies [16,20,23] reporting results on the outcome “breast cancer mortality, showed, in the fixed-effects model (*p* = 0.43), a statistically significant 14% reduction in mortality relative risk for patients enrolled versus not enrolled in MTBs (pooled HR = 0.86; 95% CI: 0.81–0.97) (Figure 2).

### 3.4. Survival

In terms of survival, the study outcomes were not unanimous. In one study, implementation of the MTB was associated with a higher five-year overall survival rate [8]. According to Brandao et al., 2021, a higher three-year overall survival rate was shown among patients with early breast cancer discussed at MTB but, with respect to three-year disease-free survival and general survival rate in patients with metastatic breast cancer, no significant differences were found between the pre-MTB and the post-MTB sub-cohorts [26]. 


*Other outcomes*


The recurrence rate, analyzed in one study only, was significantly lower for patients enrolled in an MTB than for those in a not-enrolled group [23]. 

The median time from first visit to neo-adjuvant chemotherapy for patients managed at the multidisciplinary clinic was significantly shorter than for those seen on different days by specialists from different disciplines, while the median time to definitive surgery was not significantly different between the two groups [24]. One study also reported a reduction in the use of prophylactic mastectomy following an open forum for clinical debate [17].

Furthermore, patients undergoing multidisciplinary clinical management showed faster physical recovery and better therapeutic compliance after mastectomy and breast reconstruction compared to the control group [25]. In addition, in one study, the implementation of the MTB was found to be a cost-effective measure [26]. 

From quality assessment of the studies, seven [15,17,18,19,21,22,25] were found to be of fair quality and seven [5,8,16,20,23,24,26] of good quality.

The main characteristics of the included studies are summarized in Table 1.

## 4. Discussion

In line with other findings in the literature [9,27], this study showed that the implementation of MTBs significantly impacted breast cancer care outcomes from diagnosis, to treatment, to survivorship. Out of 14 studies included, only three [20,24,26] did not show a significant association between the implementation of an MTB and outcome indicators. More specifically, they did not find a statistically significant difference between patient groups discussed or not at an MTB, respectively, with respect to mortality [20], median time to definitive surgery [24], general survival rate in patients with metastatic breast cancer or three-year disease-free survival [26]. All the other studies included found significant associations between MTB implementation and breast cancer care outcomes. As was also found in a study by Specchia and colleagues [9], the most analyzed outcomes were diagnosis, treatment and survival. Other variables impacted by the implementation of the MTBs were considered in single studies, with a high level of heterogeneity and a weaker evidence base.

In terms of diagnosis, four out of four studies [15,18,19,22], in line with others from the literature [28,29,30], showed that, with implementation of an MTB, there was a change in this outcome, especially with respect to imaging and anatomo-pathologic interpretation. 

In terms of treatment, our results also showed that, after discussion at the MTB, patients received a different recommendation from the one previously proposed by a single professional. In particular, a reduction in the use of prophylactic mastectomy was observed [17] and an increase in immediate breast reconstruction [21]. Patient survival was the most heterogeneous variable analysed, with varying outcomes observed. In some cases [8,16,23], survival rates improved after implementation of MTBs, in line with another study in the literature [28]. However, as emerged in the studies published by Rogers et al., 2017 [20] and Brandao et al., 2021 [26], observed improvements were not significant, which was in line with other studies from the literature [29,31,32]. The lack of sufficient evidence on survival, as noted by Coory et al., 2008 [32], may be attributed to the difficulty of conducting randomized clinical trials to demonstrate the potential impact of MTB on its improvement, free from confounding factors. Moreover, it is important to highlight that, of the included studies, most were retrospective in design and limited in their ability to attribute change in outcomes to multidisciplinary processes. Therefore, prospective studies could be useful to reliably assess any benefits in clinical care and care processes. 

Despite this, we were able to perform a pooled analysis of three studies [16,20,23], all of good quality, which reported results for the outcome “breast cancer mortality”. This analysis showed, in the fixed-effects model (*p* = 0.43), a statistically significant 14% reduction in mortality relative risk for patients enrolled compared to those not enrolled in MTBs. 

With respect to its weaknesses, this review relied on a relatively limited number of eligible studies, some of which had very small sample sizes. In addition, we were not able to perform a meta-analysis that included all the studies considered in the review due to the heterogeneity observed in the evaluation of different outcomes. Heterogeneity in outcomes was observed even among studies investigating the same cancer type because they involved patients with different subtypes of the cancer or who were at different stages of the same disease. These two factors could have influenced patient assessment and management. Another weakness results from the variety of studies included in the review. Most were observational studies and therefore susceptible to bias because of the study design. This was reflected in the results of the methodological assessment, which led to labeling of half of the studies as “fair”. However, although only half of the studies were labeled as “good”, none of them were found to be of “poor” quality. Finally, our review was limited to studies written in English or Italian only, and it may be that additional evidence of MTB impacts from studies written in other languages was missed.

Despite these limitations, the present review represents one of the few studies that has attempted to synthesize the research available on the topic of MTBs and their impact on healthcare outcomes, focusing on patients affected by breast cancer in a comprehensive manner and using a strict methodology. Moreover, the analysis focusing on the outcome “patient mortality” provided a strong quantitative evaluation of the topic that could represent a starting point for future investigations.

## 5. Conclusions

There is growing emphasis on the application of multidisciplinary approaches to breast cancer care. However, because of heterogeneous characterizations and contexts, the effectiveness of multidisciplinary care and of the processes that contribute to its effectiveness cannot be firmly determined. 

In terms of clinical management, this study showed that a multidisciplinary approach is a valuable way to deliver appropriate and effective care to patients affected by breast cancer, with significant gains in survival. The complex and various needs that characterize these patients’ experience require organisational improvements with enhanced coordination of care and the implementation of well-structured protocols and pathways. 

In terms of future research, further studies are needed to enrich the evidence available to date, especially with respect to outcomes which have been less investigated but which are important for the improvement of clinical care and care processes. Evidence on the impact of MTBs in clinical practice is still lacking regarding many aspects of breast cancer care. Further studies should aim to evaluate the impact on survival rates, quality of life, patient satisfaction and cost-effectiveness. The importance of the latter should not be underestimated. In a world where economic factors strongly impact the healthcare sector, the implementation of multidisciplinary teams composed of different specialists could represent a new approach to the healthcare process that could lead to improvement in diagnosis and treatment and, in turn, to cost reduction.

## Figures and Tables

**Figure 1 ijerph-19-14990-f001:**
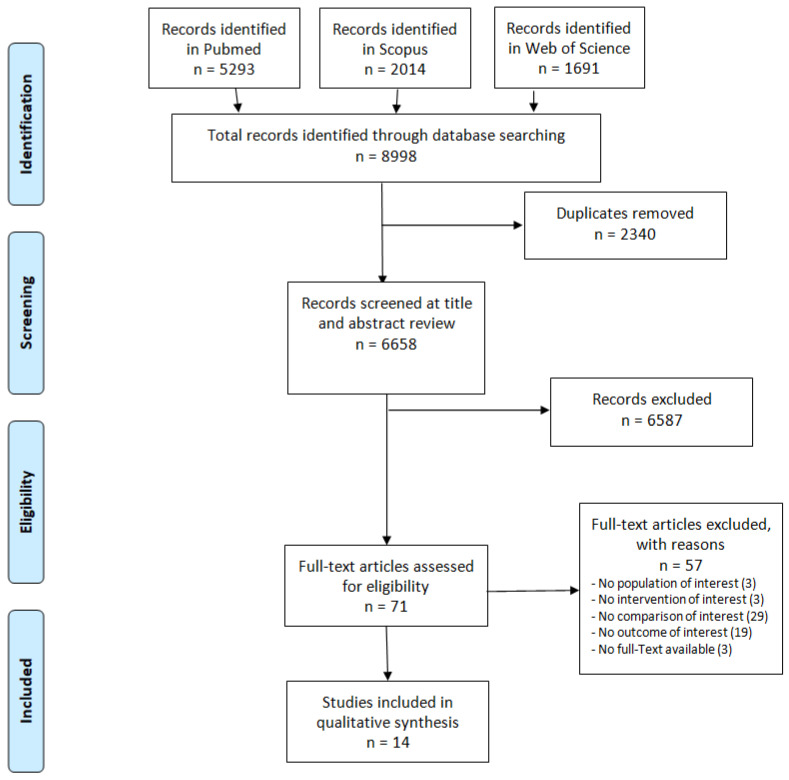
PRISMA flowchart of the included studies.

**Figure 2 ijerph-19-14990-f002:**
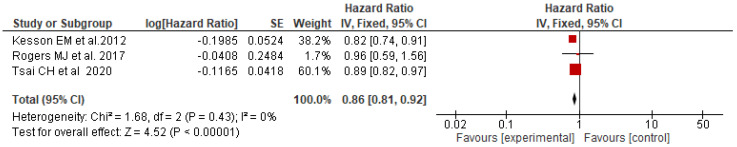
Forest plot of adjusted HRs for breast cancer mortality among patients enrolled (experimental) vs. not enrolled (control) in MTBs [16,20,23].

**Table 1 ijerph-19-14990-t001:** Characteristics of the included studies.

n	Title	Year	Country	Authors	Study Population	Study Design	Method/ Intervention	Outcome	Results	Quality Assessment
1	Changes in surgical management resulting from case review at a breast cancer multidisciplinary tumor board	2006	USA	Newman EA et al.	149 breast cancer patients	Before and after study	Retrospective review of medical records of 149 patients referred to a multidisciplinary breast cancer clinic	Diagnosis	The imaging previous interpretation was changed for 67 of 149 patients (45%). The review of the anatomo-pathologic reports resulted in a change in the findings interpretation for 43 of 149 patients (29%).	Fair
								Therapy	A change in surgical management was recommended for 77 of 149 patients (52%).	
2	Effects of multidisciplinary team working on breast cancer survival: retrospective, comparative, interventional cohort study of 13,722 women	2012	UK	Kesson EM et al.	13,722 patients diagnosed with symptomatic invasive breast cancer	Retrospective, comparative, non-randomized, interventional cohort study	Retrospective review of data from the Scottish Cancer Registry and of death records from the General Register Office for Scotland; survival comparison between intervention (introduction of multidisciplinary care) and non-intervention area	Mortality	After multidisciplinary care introduction, breast cancer mortality was 18% lower in the intervention than in the non-intervention area (adjusted HR = 0.82; 95% CI: 0.74–0.91).	Good
3	A multidisciplinary team approach minimizes prophylactic mastectomy rates	2015	UK	Leff DR et al.	151 patient requests for prophylactic mastectomy	Observational retrospective study	Retrospective review of prospectively recorded multidisciplinary team meeting records for 151 patient requests for prophylactic mastectomy (PM) from 2011 to 2014	Prophylactic mastectomy	As a result of having an open forum for clinical debate, 32.5% of requests for PM that could not be justified through inter-disciplinary discussion were declined.	Fair
4	Effect of multidisciplinary case conferences on physician decision making: breast diagnostic rounds	2016	Canada	Foster TJ et al.	76 breast cancer patients (43 with malignant diagnosis) discussed in 19 MTBs	Before and after study	Before and after diagnostic and treatment comparison between initial management plans and MTB management	Diagnosis	Of the 41% of the study sample whose treatment was changed (31 patients, 20 with malignant and 11 with benign diagnosis), 14 patients (45%) had changes in diagnostic image interpretation, and 9 patients (29%) had changes in anatomo-pathologic interpretation; in 8 (26%) patients, the change in treatment plan was attributable to both anatomo-pathologic and radiologic changes.	Fair
								Therapy	41% (31 patients) received a different recommendation from the MTB. Regarding modified treatment recommendations, in 9% of cases, the patient was advised against immediate surgery; in 7% of cases, invasive surgery replaced previously planned decision; in 5% of cases, the recommended surgery was modified.	
5	Practice and impact of multidisciplinary tumor boards on patient management: A Prospective Study	2016	Lebanon	Charara RN et al.	503 cases presented at MTBs at the American University of Beirut Medical Center (AUBMC) between October 2013 and December 2014; 149 (29.6%) of cases presented at breast cancer MTBs.	Before and After Study	Evaluation of change in plans resulting from MTB discussions	Diagnosis	2.7% of breast cancer cases had a change in diagnosis.	Fair
								Therapy	Plans for new treatment were made in 67% of breast cancer cases.	
6	Comparison of outcomes for cancer patients discussed and not discussed at a multidisciplinary meeting	2017	Australia	Rogers MJ et al.	657 breast cancer patients; 366 (56%) presented to MTB within 60 days after diagnosis; 409 (62%) overall	Historical cohort study	Retrospective analysis of data from a single center for patients discussed and not discussed by MTB	Mortality	No statistically significant difference in mortality was found between patients discussed and patients not discussed (adjusted HR = 0.96, 95% CI: 0.59–1.57).	Good
								Therapy	Breast cancer patients discussed at an MTB were found to undergo surgery either alone or with a systemic agent or all treatment types significantly more than patients not discussed (*p* < 0.01).	
7	Improved immediate breast reconstruction as a result of oncoplastic multidisciplinary meeting	2017	UK	El Gammal MM et al.	229 breast cancer patients (120 mastectomy discussed; 109 mastectomy not discussed)	Before and After Study	Retrospective analysis performed comparing patients managed before and after the introduction of a MTB	Therapy	Before MTB introduction, among 109 patients who underwent mastectomy, 31 (28%) had breast reconstruction and 64% of the latter (20/31) had immediate breast reconstruction (IBR). After introducing oncoplastic MTB, among 120 women who underwent mastectomy, 50 (42%) had breast reconstruction and 39 (78%) of the latter had IBR. The increase in reconstruction after introduction of the oncoplastic MTB was statistically significant (*p* = 0.0144).	Fair
8	The value of a second opinion for breast cancer patients referred to a National Cancer Institute (NCI)-designated cancer center with a multidisciplinary breast tumor board	2018	USA	Garcia D et al.	70 breast cancer patients seeking second opinions	Before and after study	Comparison of diagnostic data for the same patients before and after discussion at the MTB	Diagnosis	After MTB discussion, for 43 patients (61%) further diagnostic imaging or biopsy (33 additional imaging and 30 additional biopsies) were required. Among them, 16 (23% of the total sample) had a previously undiagnosed metastasis. The second opinion led to a change in the interpretation of pathology for 14 of them (20%). 11 of 70 patients were sent for genetic counselling which had not been required before the MTB; in 2 of these, 11 patient mutations of uncertain significance were found, which did not change disease management.	Fair
9	Factors associated with multidisciplinary consultations in patients with early-stage breast cancer	2019	USA	Quyyumi FF et al.	61,039 patients, 43,280 of which with multidisciplinary care, aged 65 and older, diagnosed with stages I–III breast cancer, who underwent breast-conserving surgery (BCS) within 6 months of their diagnosis	Historical cohort study	Assessment of the association between multidisciplinary care and nationally recognized quality indicators in patients with breast cancer	Therapy	The implementation of multi-disciplinary care was associated with an increased likelihood of meeting three nationally recognized quality indicators: adjuvant hormone therapy for hormone-receptor-positive tumors, chemotherapy for hormone-receptor-negative cancer, and radiation after lumpectomy.	Good
10	The improved effects of a multidisciplinary team on the survival of breast cancer patients: experiences from China	2019	China	Lu J et al.	16354 patients undergoing breast cancer surgery during the period 2006–2016 at the Fudan University Shanghai Cancer Center (299 patients treated by a MTB and 16,055 patients not treated by a MTB)	Historical cohort study	Retrospective analysis of data from a single center for patients discussed and not discussed by MTB	Survival	The five-year survival rate of breast cancer patients discussed by a well-organized MTB was 15.6% higher than that of women not discussed by MTB. Survival time was longer for patients in the well-organized MDT group than patients in the non-MDT group (HR = 0.4, *p* = 0.014).	Good
11	Effect of multidisciplinary team care on the risk of recurrence in breast cancer patients: A national matched cohort study	2020	Taiwan	Tsai CH et al.	9266 breast cancer patients enrolled in MTB vs. 9266 not enrolled	Historical cohort study	Comparison of two cohorts of breast cancer patients, respectively enrolled and not enrolled in MTB	Recurrence	The recurrence rate was significantly lower for patients enrolled in MTB (HR = 0.84; 95% CI: 0.70–0.99; *p* < 0.05) than for the not-enrolled group.	Good
								Mortality	The relative risk of mortality was significantly lower for patients enrolled in MTB than for the not-enrolled group (adjusted HR = 0.89; 95% CI: 0.82–0.96)	
12	The effect of 1-day multidisciplinary clinic on breast cancer treatment	2020	USA	Akhtar Z et al.	296 patients who were treated at Johns Hopkins for stage II or III breast cancer between May 2015 and December 2017; 220 of them (74%) were seen at interdisciplinary clinic (IDC), i.e., on different days by specialists from varying disciplines, and 76 (24%) in a new single-day multidisciplinary clinic (MDC) with coordination between two or three specialties	Historical cohort study	Retrospective analysis of data from a single center for patients managed at MDC vs. ICD	Time from first visit to NACT (neoadjuvant chemotherapy)	The median time from first visit to NACT was shorter for patients managed at the MDC (13 days) than for those seen at the IDC (22 days) (HR = 3.5; 95% CI: 1.8–6.9; *p* < 0.001).	Good
								Time from first visit to surgery	The median time to definitive surgery was not significantly different between two groups (32 days for MDC; 31 days for ICD; HR = 1.2; 95% CI: 0.72–2.0; *p* = 0.47).	
13	A multidisciplinary approach to breast cancer introducing a management file for breast cancer patients	2020	Romania	Baciu AC et al.	20 breast cancer patients divided into 2 groups: in the study group (10 patients) the multidisciplinary management file for breast cancer patients (MMFBCP) was used, while in the control group (10 patients), the classic interdisciplinary approach (IDC) was applied	Cohort study	Prospective study on 10 MMFBCP vs. 10 IDC breast cancer patients	Therapy	Six patients in the study group had breast reconstruction vs. 3 in the control group.	Fair
								Physical recovery after surgery	Patients enrolled in the study group showed faster physical recovery after mastectomy and breast reconstruction than those enrolled in the control group.	
								Therapeutic compliance	Patients enrolled in the study group showed better post-mastectomy and breast reconstruction therapeutic compliance than those enrolled in the control group.	
14	Survival impact and cost-effectiveness of a multidisciplinary tumor board for breast cancer in Mozambique, Sub-Saharan Africa	2021	Mozambique	Brandao M et al.	205 patients with breast cancer diagnosed (98 before and 107 after MTB implementation)	Historical cohort study	Retrospective analysis of data from a single center for patients discussed and not discussed by MTB	Survival	With respect to the 3-year overall survival, the adjusted HR for death in the post-MTB sub-cohort vs. the pre-MTB one was 0.77; 95% CI: 0.49–1.19. Among patients with early breast cancer, the adjusted HR for death in the post-MTB vs. the pre-MTB sub-cohort was 0.47; 95% CI: 0.27–0.81. For patients with metastatic breast cancer, no survival benefit was observed with the introduction of the MTB.	Good
								Disease-free survival	With respect to 3-year disease-free survival, there were no significant differences between the pre-MTB and the post-MTB sub-cohorts (adjusted HR for relapse or death = 0.72; 95% CI: 0.46–1.13).	
								MTB cost-effectiveness	The implementation of MTB was found to be a cost-effective measure (ICER = $802.96 per QALY).	

## Data Availability

Not applicable.

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
