# Peer review of "The Impact of Tumor Boards on Breast Cancer Care: Evidence from a Systematic Literature Review and Meta-Analysis"

_ijerph, 2022, doi:10.3390/ijerph192214990_

Round 1

Reviewer 1 Report

Thanks for the chance to review. The following would improve this manuscript:

Add rationale for 1995 search window

More detail needed on the PICO criteria - 

-how was "multidisciplinary" defined?

-"lack of MTB implementation" - how was this defined? Did you accept studies with no comparison group? What about comparative effectiveness studies? Nonrandomized studies? Were any study designs excluded (e.g. qualitative, cross sectional)? What are examples of clinical and process outcomes? 

How did you determine which data could be pooled for meta-analysis? (e.g. measures of 5 year and 3-year survival)

Author Response

- Add rationale for 1995 search window

A rationale and the reference for the search window have been added in lines 94-97

More detail needed on the PICO criteria:

- How was "multidisciplinary" defined?

The definition and the reference have been added in lines 105-107

- "lack of MTB implementation" - how was this defined? Did you accept studies with no comparison group? What about comparative effectiveness studies? Nonrandomized studies? Were any study designs excluded (e.g. qualitative, cross sectional)? What are examples of clinical and process outcomes?

  • The “lack of MTB implementation” was intended as patients’ clinical management without the intervention of multidisciplinary teams (see lines 108-109).
  • In regards to the study design of the included studies, as stated in the materials and methods paragraph, only reviews, letters and commentaries were excluded because not reporting original data. The search has been open to any possible result that followed the stated inclusion/exclusion criteria. We reported in Table 1 for each study and in the Results paragraph (lines 163-166) the study designs of the included studies (6 historical cohort studies, 5 before and after studies, 1 cohort study, 1 retrospective interventional cohort study and 1 observational retrospective study). All the studies had a comparison group.
  • Taking into account the explanation given at lines 109-111, examples of clinical outcomes emerged in our results are survival, mortality and recurrence rate. Examples of process outcomes are: time from visit to chemotherapy/ first surgery, changes in diagnosis and therapy.

- How did you determine which data could be pooled for meta-analysis? (e.g. measures of 5 year and 3-year survival)

Thank you for your comment, useful to better explain this concept in the paper. Your doubt may be due to the fact that we have summarized in the same paragraph two different, even though related, concepts: mortality and survival. The metanalysis could have been done only for the mortality as long as the 3 articles described mortality in similar ways, while survival has been evaluated by 3 articles in non-comparable way.

In order to make it clearer, in the results paragraph ‘3.3 Survival’ has been split into ‘3.3 Mortality’ and ‘3.4 Survival’ (lines 193-228); outcomes in Table 1 has been updated and a sentence has been added in the methodology section (lines 138-139). Changes have been also made in the Results paragraph (lines 168-172).

Reviewer 2 Report

This manuscript by Di Pilla et al, represents a systematic review and meta- analysis regarding the impact of multidisciplinary meetings on breast cancer care. This is indeed a subject of paramount importance as there appears to be growing evidence that patients treated within units that incorporate tumor boards within their approach have better outcomes regarding their cancer prognosis and survival. This manuscript highlights important issues in breast cancer treatment, such as the impact of a tumor board on the decision for surgery, or the period of time until initiation of neoadjuvant treatment.

Comments to the authors:

1. The authors could include the following reference in the introduction supporting the importance of a tumor board in a specialized breast unit. Bingazoli et al, The Breast 2020, 51:65-84.

2. It would be advisable to register the meta-analysis protocol within the PROSPERO platforma.

Author Response

Comments to the authors:

  1. The authors could include the following reference in the introduction supporting the importance of a tumor board in a specialized breast unit. Bingazoli et al, The Breast 2020, 51:65-84.

Thank you for the suggestion. The above has been added as reference n°28. The following references’ numbers have been changed also in the text.

  1. It would be advisable to register the meta-analysis protocol within the PROSPERO platform.

Thank you for the valuable and appropriate suggestion. Unfortunately, since October 2019 PROSPERO accepts reviews for which data extraction have not been started yet. At this stage it is therefore not possible to register the meta-analysis protocol. We will surely take it into account for future studies.

Round 2

Reviewer 2 Report

The authors have addressed my comments.